# L-shaped relationship between hemoglobin glycation index and short-term mortality in patients with intracerebral hemorrhage: A retrospective cohort study

Jia Xu[1], Guangdong Wang[2], Yongfeng Ni[3]*

**1** Department of Electrocardiography Diagnosis, The Second Affiliated Hospital of Anhui Medical University, Hefei, Anhui, China, **2** Department of Respiratory and Critical Care Medicine, First Affiliated Hospital of Xi'an Jiaotong University, Xi'an, Shanxi, China, **3** Department of Neurosurgery, Anqing First People's Hospital of Anhui Medical University, Anqing, Anhui, China

\* ahmunyf@126.com

## Abstract

### Background

Intracerebral hemorrhage (ICH) carries a substantial risk of early death and is frequently linked to unfavorable clinical outcomes, yet early prognostication remains challenging. The hemoglobin glycation index (HGI), which quantifies the discrepancy between measured glycosylated hemoglobin A1c(HbA1c) and the level expected based on fasting plasma glucose (FPG), has shown prognostic relevance in various clinical settings. Our objective was to investigate whether the HGI serves as a predictor of short-term mortality among individuals with ICH.

### Methods

We performed a retrospective analysis utilizing data from the critical care database. We identified 1,318 adult ICH patients with available HbA1c and FPG data. HGI was defined as the difference between the observed HbA1c level and the HbA1c value predicted on the basis of admission FPG. Patients were stratified into HGI tertiles. The main outcome was 30-day all-cause mortality (90-day mortality served as a secondary endpoint). The relationship between HGI and mortality was examined using Kaplan–Meier curves, Cox proportional hazards models, and restricted cubic spline analyses. Subgroup analyses were also performed.

### Results

Mortality was significantly greater among individuals in the lowest HGI tertile than among those in the highest tertile (30.18% vs 17.24% at 30-day mortality, p < 0.001; 33.64% vs 22.07% at 90-day mortality, p < 0.001). Multivariable Cox regression showed that higher HGI was an independent predictor of reduced mortality risk. The

**Data availability statement:** Data are available from the MIMIC-IV database, accessible through https://mimic.mit.edu/.

**Funding:** The author(s) received no specific funding for this work.

**Competing interests:** The authors have declared that no competing interests exist.

adjusted hazard ratio(HR) for 30-day mortality was 0.84 (95% CI 0.75–0.93), for 90-day mortality was 0.86(95% CI: 0.79–0.95) in fully adjusted models. Restricted cubic spline (RCS) analysis demonstrated an L-shaped association, with inflection points identified at HGI values of 0.692 for 30-day mortality and 0.472 for 90-day mortality, respectively. Below these thresholds, each one-unit increase in HGI corresponded to a 47.7% reduction in 30-day mortality risk and a 40.4% reduction in 90-day mortality risk, respectively. The association between HGI and mortality was consistent across most subgroups, with a significant interaction by diabetes status (p for interaction < 0.05), indicating the predictive value of HGI was more pronounced in patients with diabetes.

## Conclusion

An L-shaped association exists between HGI and short-term mortality in ICH patients, with low HGI indicating substantially higher risk and holds potential as a novel prognostic indicator for facilitating early risk stratification in patients with acute ICH.

## Introduction

Intracerebral hemorrhage (ICH) constitutes a highly lethal form of stroke, defined by non-traumatic extravasation of blood into the cerebral parenchyma [1]. Although it accounts for a smaller proportion of total stroke cases, it contributes disproportionately to stroke-related mortality and morbidity [2]. Despite advances in acute stroke care, ICH outcomes remain poor: the 30-day mortality rate approaches 40–50%, and only a small proportion of patients regain meaningful neurological function [3–5]. The condition imposes a substantial economic and caregiver burden due to prolonged hospitalization, intensive care requirements, and high rates of long-term disability. These factors underscore the critical need for improved prognostic tools and early risk stratification strategies to guide clinical management and resource allocation in ICH care. glycated hemoglobin

The hemoglobin glycation index (HGI) is a novel indicator of glycemic discrepancy that captures the interindividual variation between observed glycosylated hemoglobin A1c(HbA1c) and the value estimated from fasting plasma glucose (FPG) [6]. First introduced in the early 2000s, HGI reflects an individual's propensity for non-enzymatic hemoglobin glycation beyond what is expected from ambient glycemia alone [7,8]. This index has been hypothesized to reflect intrinsic biological differences in protein glycation and red blood cell turnover, with potential implications for vascular health and inflammatory processes. Given its potential link to endothelial dysfunction, vascular damage, and inflammatory pathways, HGI has attracted significant interest as a possible marker for cardiovascular diseases and other conditions associated with vascular integrity [9–11]. Despite the growing interest in HGI as a prognostic marker, investigation of its utility in forecasting clinical outcomes among patients

with ICH has only recently emerged. A concurrent study by Sun et al. [12] demonstrated an association between HGI and 30-day mortality in ICH patients, identifying a threshold effect at an HGI of 0.78. Building upon these foundational findings, our study aims to extend the current understanding by expanding the outcome scope beyond acute-phase mortality, exploring potential time-dependent shifts in prognostic thresholds, and investigating effect modification by clinical factors such as diabetes status. These endeavors may inform more nuanced risk stratification strategies for this clinically vulnerable population.

This analysis aims to investigate how variations in the HGI relate to short-term mortality in patients with ICH, based on Medical Information Mart for Intensive Care(MIMIC)-IV database [13]. Given the complexity and variability in the clinical course of ICH, understanding how HGI correlates with patient outcomes could inform personalized treatment strategies and improve prognostic accuracy. This work, therefore, aims to add to the expanding evidence base regarding the use of HGI in short-term prognosis of ICH patients, and its potential to enhance clinical decision-making in ICH management.

## Methods

### Data source

We utilized the MIMIC-IV (version 3.1) critical care database, which contains de-identified health records from intensive care units (ICUs). This dataset covers a wide range of patient-related information, including demographics, vital signs, laboratory results, treatments, diagnoses, and clinical outcomes, making it an invaluable resource for researchers investigating critical care, machine learning models for healthcare, and predictive analytics in ICU settings. The database is publicly available to credentialed researchers and has been curated to ensure patient privacy and data quality. Access was obtained through a data use agreement and completion of required training. One author (Jia Xu, Certification ID 64822128) had approved access to MIMIC-IV and performed the data extraction for this study. All relevant data were queried and retrieved using structured query language (SQL).

### Study population, data extraction, and definition of HGI

All ICU patients in the MIMIC-IV database documented with ICH were selected based on corresponding ICD-9 or ICD-10 classification codes. This query yielded 3,148 ICH admissions. We then applied exclusion criteria: individuals under the age of 18(n = 0), individuals without an available HbA1c measurement (n = 1,809), and patients without a recorded FBG (n = 21) were excluded. After these exclusions, the final analysis cohort comprised 1,318 adult ICH patients for final analysis. To assess potential selection bias, we conducted a comparative analysis of baseline characteristics, clinical interventions, and outcomes between patients with missing HbA1c measurements (n = 1,809) and those included in the final cohort (n = 1,318).

For each included patient, we extracted clinical variables across several domains: demographics, baseline clinical severity scores, vital signs, comorbidities, laboratory measurements, and treatments. Table 1 summarizes the variables collected. To ensure consistency, measurements of physiological parameters, clinical scores, and biochemical indices obtained during the initial 24 hours following ICU admission were included. Treatment-related information was extracted from the ICU records during the patient's stay. Data with more than 10% missingness were excluded (S1 Table), while the remaining incomplete entries were addressed using multiple imputation techniques.

HGI was calculated in two steps. First, we derived a predicted HbA1c for each patient based on their FBG using a linear regression model built on the study cohort. This regression yielded the formula: predicted HbA1c = 0.013 * FPG + 4.354, with a correlation coefficient r = 0.594 (P < 0.001), indicating a moderate positive correlation between FBG and HbA1c in our cohort (Fig 1). Second, each patient's HGI was derived by subtracting the HbA1c value estimated using this formula from the observed HbA1c measurement [14]. Thus, a positive HGI indicates that the observed HbA1c is higher than predicted from ambient glucose levels (i.e., more non-enzymatic glycation than expected), whereas a negative

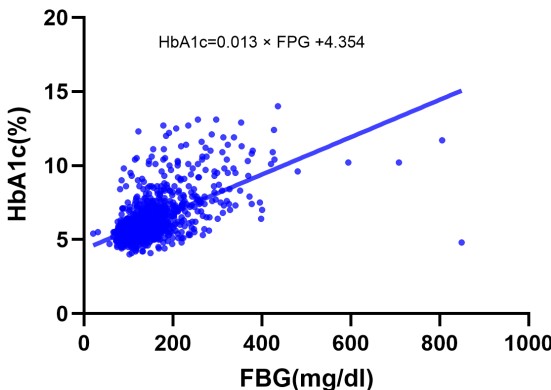 PLOS One

**Table 1. The detailed extracted variables in MIMIC-IV database.**

| Items | Composition |
|---|---|
| Demographic information | age; gender; race |
| Severity scores | SOFA; GCS |
| Vital signs | Heart Rate; SBP; DBP; respiratory rate; SpO2 |
| Comorbidities | hypertension; diabetes; myocardial infarction; congestive heart failure; peripheral vascular disease; chronic pulmonary disease; AKI; sepsis |
| Laboratory test results | WBC; Hemoglobin; platelet; BUN; creatinine; potassium; sodium; FBG; HbA1c; anion gap; PT; INR; |
| Treatments | mannitol; heparin; warfarin; vasoactive drug; beta blockers; diuretic; insulin; ventilation; cerebral surgery |

Abbreviations: SOFA, sequential organ failure assessment; GCS, Glasgow coma scale; SBP, Systolic Blood Pressure; DBP, Diastolic Blood Pressure; SpO2, oxygen saturation; AKI, acute kidney injury; WBC, white blood cell; BUN, blood urea nitrogen; FBG, fasting blood glucose; HbA1c, Hemoglobin A1c; PT, prothrombin time; INR, international normalized ratio; ICU, intensive care unit.

HbA1c=0.013 × FPG +4.354

**Fig 1. The linear correlation between FBG and HbA1c levels.** FBG, fasting blood glucose; HbA1c, Hemoglobin A1c.

HGI indicates that the observed HbA1c is lower than predicted, reflecting less glycation than expected based on the prevailing glucose concentration.

Patients were subsequently stratified according to HGI tertiles as follows: Tertile(T)1 (HGI < −0.481, n = 434), T2 (−0.481 ≤ HGI < 0.066, n = 449), and T3 (HGI ≥ 0.066, n = 435). A schematic diagram outlining the selection process, as well as HGI-based classification, is shown in Fig 2.

## Clinical outcome events

The primary endpoint evaluated was death from any cause within 30 days of hospital admission, whereas the secondary endpoint considered mortality occurring within 90 days post-admission.

## Statistical analysis

Categorical data were summarized as frequencies and proportions and comparisons across HGI tertiles were conducted using the chi-squared test. Continuous data were summarized as medians with interquartile ranges (IQR) after assessment for normality, and differences among groups were evaluated using nonparametric rank-sum test. Survival analysis

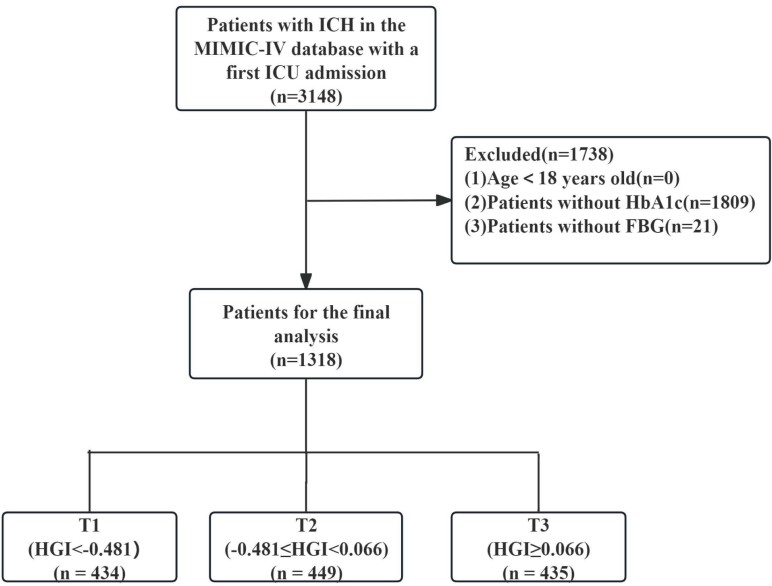

**Fig 2. Flow chart of population selection.** ICH, Intracerebral hemorrhage; MIMIC, Medical Information Mart for Intensive Care IV; ICU, intensive care unit; FBG, fasting blood glucose; HbA1c, Hemoglobin A1c; HGI, hemoglobin glycation index.

based on HGI tertiles was conducted via the Kaplan-Meier approach, and intergroup differences were evaluated using the log-rank test. The association between HGI and clinical outcomes was quantified by calculating hazard ratios (HRs) using the Cox proportional hazards model. To account for potential confounders, three hierarchical multivariate Cox regression models were constructed: Model one was crude; Model two accounted for adjusted for baseline demographics and vital signs. Model three adjusted for Model 2 plus clinical severity scores and comorbidities, such as sequential organ failure assessment (SOFA), Glasgow coma scale (GCS), myocardial infarction, sepsis, congestive heart failure, white blood cell (WBC), hemoglobin, platelet, creatinine, blood urea nitrogen (BUN), potassium levels, along with treatment factors such as mannitol, diuretic, insulin, vasoactive drug, ventilation, cerebral surgery. To examine the association between HGI and clinical outcomes in greater detail, restricted cubic spline (RCS) modeling was utilized, and a piecewise regression analysis was conducted to determine the HGI inflection point. Additionally, stratified analyses were performed to evaluate the robustness and generalizability of the findings across key subgroups. Statistical analyses were conducted using R Studio, with significance defined as a two-tailed P value less than 0.05.

## Results

### Baseline characteristics

This study involved 1318 patients with ICH, presenting a median age of 72 years (IQR: 60−82). Of this cohort, 616(46.74%) were female, and 531(40.29%) were non-white (Table 2). The all-cause mortality rates were 21.09%(n = 278) at 30 days and 25.64%(n = 338) at 90 days. Individuals were stratified into tertiles according to HGI. The median values of HGI for these tertiles were −0.8 (IQR: −1.1, −0.6), −0.2 (IQR: −0.4, −0.1), and 0.5 (IQR: 0.2, 1.3), respectively.

Table 2 summarizes the baseline information and clinical outcome events for each group. Relative to patients in the highest HGI tertile, those in the lowest tertile exhibited a smaller proportion of males, along with reduced levels of hemoglobin, platelet count, BUN, and potassium. However, the T1 group exhibited higher heart rate, SOFA score, WBC counts,

**Table 2. Characteristics and outcomes of participants categorized by HGI.**

| Characteristics | Total (n = 1318) | T1 (HGI < −0.481) (n = 434) | T2 (−0.481 ≤ HGI < 0.066) (n = 449) | T3 (HGI ≥ 0.066) (n = 435) | P-value |
|---|---|---|---|---|---|
| HGI | −0.2 (−0.6, 0.2) | −0.8 (−1.1, −0.6) | −0.2 (−0.4, −0.1) | 0.5 (0.2, 1.3) | < 0.001 |
| Age (year) | 72(60, 82) | 71 (58, 81) | 73 (61, 83) | 71(62, 82) | 0.185 |
| Gender, n (%) | | | | | 0.023 |
| Female | 616 (46.74) | 225 (51.84) | 205 (45.66) | 186 (42.76) | |
| Male | 702 (53.26) | 209 (48.16) | 244 (54.34) | 249 (57.24) | |
| Race, n (%) | | | | | 0.488 |
| Non-White | 531 (40.29) | 168 (38.71) | 178 (39.64) | 185 (42.53) | |
| White | 787 (59.71) | 266 (61.29) | 271 (60.36) | 250 (57.47) | |
| **Vital signs** | | | | | |
| Heart rate (beats/min) | 81 (70, 92) | 83(72, 95) | 81 (71, 92) | 78 (69, 91) | < 0.001 |
| SBP (mmHg) | 138 (125, 151) | 137(122, 151) | 137 (126, 149) | 140(125, 153) | 0.293 |
| DBP (mmHg) | 77 (66, 88) | 77(65, 88) | 77 (65, 88) | 75 (66, 880) | 0.779 |
| Respiratory rate (times/min) | 18 (16, 22) | 18 (16, 22) | 18 (15, 21) | 18 (16, 21) | 0.322 |
| SpO2 (%) | 98 (96, 100) | 98(96, 100) | 97 (95, 99) | 98 (96, 99) | 0.006 |
| **Severity scores** | | | | | |
| SOFA | 3.00 (1.00, 4.00) | 3.00 (2.00,4.75) | 2.00 (1.00,3.00) | 2.00 (1.00,4.00) | < 0.001 |
| GCS | 14.00 (12.00, 15.00) | 14.00 (11.00,15.00) | 14.00 (12.00,15.00) | 14.00 (12.00,15.00) | 0.824 |
| **Comorbidity, n (%)** | | | | | |
| Congestive heart failure | 226 (17.15) | 59 (13.59) | 70 (15.59) | 97 (22.30) | 0.002 |
| Hypertension | 1114 (84.52) | 359 (82.72) | 368 (81.96) | 387 (88.97) | 0.007 |
| Diabetes | 407 (30.88) | 73 (16.82) | 76 (16.93) | 258 (59.31) | < 0.001 |
| Chronic Pulmonary Disease | 144 (10.93) | 58 (13.36) | 35 (7.80) | 51 (11.72) | 0.193 |
| AKI | 1001 (75.95) | 345 (79.49) | 325 (72.38) | 331 (76.09) | 0.047 |
| Myocardial Infarction | 121 (9.18) | 35 (8.06) | 31 (6.90) | 55 (12.64) | 0.008 |
| Peripheral Vascular Disease | 86 (6.53) | 18 (4.15) | 30 (6.68) | 38 (8.74) | 0.023 |
| Sepsis | 452 (34.29) | 164 (37.79) | 145 (32.29) | 143 (32.87) | 0.171 |
| **Laboratory parameters** | | | | | |
| WBC (K/μL) | 9.90 (7.82, 12.60) | 10.80 (8.40,14.00) | 9.70 (7.70,12.10) | 9.50 (7.50,12.00) | < 0.001 |
| Hemoglobin(g/dl) | 12.6 (11.2, 13.8) | 12.2 (10.8, 13.8) | 12.8 (11.6, 14.1) | 12.7 (11.4, 13.7) | < 0.001 |
| Platelet (K/μL) | 209.00 (168.00, 260.75) | 201.00 (158.25,257.00) | 214.00 (175.00,265.00) | 210.00 (168.50,261.50) | 0.011 |
| BUN (mg/dL) | 16.00 (12.00, 22.00) | 16.00 (12.00,22.00) | 16.00 (12.00,20.00) | 17.00 (13.00,23.00) | 0.003 |
| Creatinine (mg/dL) | 0.90 (0.70, 1.10) | 0.90 (0.70,1.10) | 0.90 (0.70,1.10) | 0.90 (0.80,1.20) | 0.012 |
| Potassium (mmol/L) | 4.00 (3.70, 4.30) | 3.90 (3.60,4.30) | 4.00 (3.70,4.20) | 4.00 (3.70,4.40) | 0.036 |
| Sodium (mmol/L) | 139.00 (137.00, 142.00) | 139.00 (137.00,142.00) | 140.00 (138.00,142.00) | 139.00 (137.00,142.00) | 0.191 |
| Aniongap (mmol/L) | 14.00 (12.00, 16.00) | 14.00 (12.00,17.00) | 14.00 (12.00,16.00) | 14.00 (12.00,16.00) | 0.004 |
| PT(s) | 12.50 (11.60, 13.80) | 12.40 (11.60,13.97) | 12.40 (11.70,13.70) | 12.60 (11.70,14.05) | 0.136 |
| INR | 1.10 (1.10, 1.30) | 1.10 (1.10,1.30) | 1.10 (1.10,1.20) | 1.20 (1.10,1.30) | 0.203 |
| **Treatment, n (%)** | | | | | |
| Mannitol | 142 (10.77) | 65 (14.98) | 38 (8.46) | 39 (8.97) | 0.003 |
| Heparin | 876 (66.46) | 315 (72.58) | 286 (63.70) | 275 (63.22) | 0.004 |
| Warfarin | 38 (2.88) | 6 (1.38) | 14 (3.12) | 18 (4.14) | 0.049 |
| Beta_blockers | 572 (43.40) | 172 (39.63) | 203 (45.21) | 197 (45.29) | 0.154 |
| Diuretic | 417 (31.64) | 146 (33.64) | 124 (27.62) | 147 (33.79) | 0.078 |

*(Continued)*

**Table 2.** (Continued)

| Characteristics | Total (n = 1318) | T1 (HGI < −0.481) (n = 434) | T2 (−0.481 ≤ HGI < 0.066) (n = 449) | T3 (HGI ≥ 0.066) (n = 435) | P-value |
|---|---|---|---|---|---|
| Insulin | 1009 (76.6) | 338 (77.9) | 321 (71.5) | 350 (80.5) | 0.005 |
| Vasoactive drug | 142 (10.77) | 51 (11.75) | 42 (9.35) | 49 (11.26) | 0.477 |
| Ventilation | 837 (63.51) | 297 (68.43) | 269 (59.91) | 271 (62.30) | 0.026 |
| Cerebral Surgery | 105 (7.97) | 54 (12.44) | 25 (5.57) | 26 (5.98) | < 0.001 |
| **Outcomes** | | | | | |
| Hospital stay (day) | 7.88 (4.38, 15.58) | 8.27 (4.69,15.12) | 7.68 (4.50,13.65) | 7.90 (4.01,16.81) | 0.572 |
| ICU stay (day) | 3.96 (1.89, 8.17) | 4.75 (2.24,8.86) | 3.71 (1.84,7.78) | 3.63 (1.71,8.16) | 0.006 |
| Hospital mortality, n (%) | 215 (16.31) | 107 (24.65) | 47 (10.47) | 61 (14.02) | < 0.001 |
| ICU mortality, n (%) | 145 (11.00) | 72 (16.59) | 35 (7.80) | 38 (8.74) | < 0.001 |
| 30-day mortality, n (%) | 278 (21.09) | 131 (30.18) | 72 (16.04) | 75 (17.24) | < 0.001 |
| 90-day mortality, n (%) | 338 (25.64) | 146 (33.64) | 96 (21.38) | 96 (22.07) | < 0.001 |

Abbreviations: HGI, hemoglobin glycation index; SBP, systolic blood pressure; DBP, diastolic blood pressure; SpO2, oxygen saturation; SOFA, sequential organ failure assessment; GCS, glasgow coma scale; AKI, acute kidney injury; WBC, white blood cell; RDW, red cell distribution width; BUN, blood urea nitrogen; PT, prothrombin time; HDL, high density lipoprotein; INR, international normalized ratio; ICU, intensive care unit.

high-density lipoprotein and cholesterol levels. The percentages of myocardial infarction, congestive heart failure (CHF), hypertension, diabetes, and peripheral vascular disease were lower in T1 group than T3 group, while the incidence of acute kidney injury (AKI) was higher in T1. Additionally, a higher frequency of mannitol, heparin, mechanical ventilation, and cerebral surgery was observed in T3, while insulin use was less frequent. The mortality rates in hospital and ICU were 16.31% (n = 215) and 11%(n = 145), respectively. The T1 group had higher all-cause mortality rates than the T3 group in both the hospital and ICU settings.

The survival outcomes among patients were assessed through Kaplan-Meier survival analysis stratified by HGI tertiles (Fig 3). The T1 group showed markedly elevated mortality at both 30-day and 90-day follow-ups relative to the other groups (30.18% vs. 16.04%, and 17.24% at 30 days; 33.64% vs. 21.38%, and 22.07 at 90 days; all log-rank P < 0.001). The results indicate that a low HGI correlates with poorer short-term prognosis in ICH patients.

## Comparison of baseline characteristics between patients with missing HbA1c measurements and included ICH patients

S2 Table presents the comparison between patients with missing HbA1c measurements (n = 1,809) and those included in the final cohort (n = 1,318). The two groups were well balanced in core prognostic factors for ICH, including GCS score, gender distribution, heart rate, history of myocardial infarction, and mechanical ventilation use, indicating comparable neurological severity and critical illness status between groups.

Statistically significant differences were observed in age, diabetes prevalence, insulin use, SOFA score, mortality, and several treatments. The included group was older and had a higher prevalence of diabetes. The excluded group had slightly higher SOFA scores and higher mortality. Treatment differences were also noted: the excluded group more frequently received mannitol, vasoactive drugs, and cerebral surgery, while the included group had higher rates of heparin, insulin, and diuretics.

## The association between HGI and clinical outcome events

S3 Table presents the baseline disparities according to 30-day survival status. Survivors demonstrated a significantly higher HGI than non-survivors (−0.19 vs. −0.44, P < 0.001). In contrast, patients who did not survive

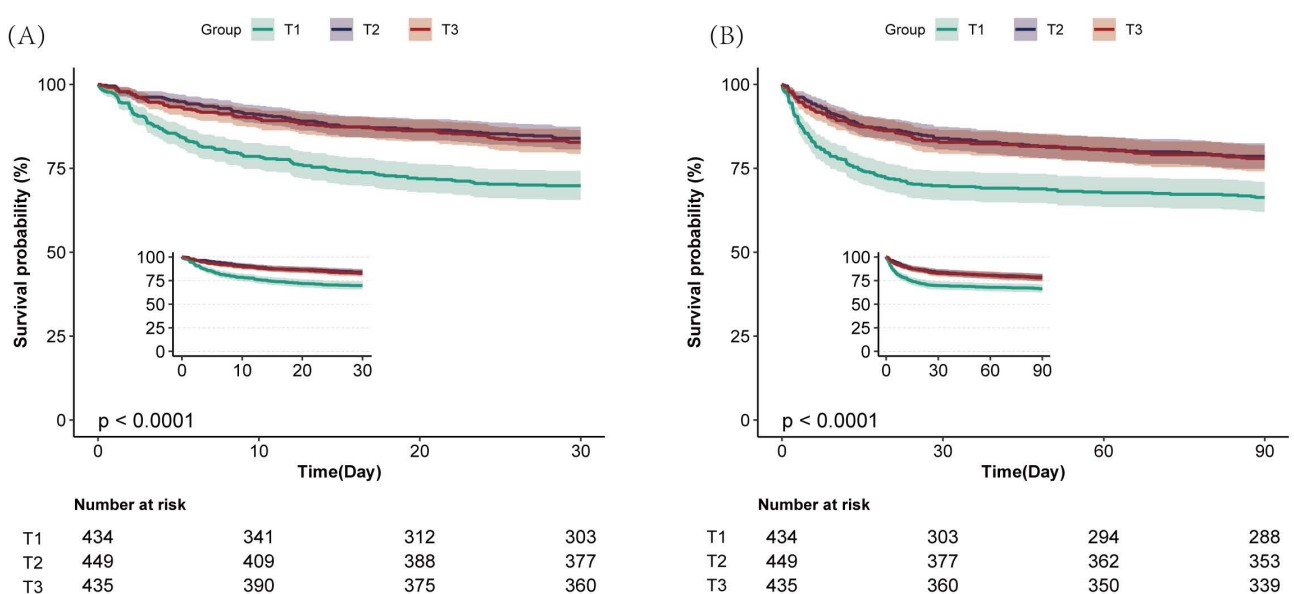

**Fig 3. Kaplan–Meier survival analysis curves for all-cause mortality at 30-day (A) and 90-day (B) across different HGI tertiles.** HGI, hemoglobin glycation index.

tended to be older, female, non-white and had higher respiratory rate, oxygen saturation (SpO2) levels, and SOFA scores. In addition, they showed a greater incidence of CHF, myocardial infarction and sepsis. Laboratory findings revealed elevated WBC, hemoglobin, BUN, creatinine, potassium, prothrombin time, and international normalized ratio levels among non-survivors, along with decreased platelet counts. Treatment-wise, non-survivors were more often treated with mannitol, diuretics, insulin, vasoactive drugs, mechanical ventilation, and underwent cerebral surgery.

Univariate Cox proportional analysis for 30-day mortality was presented in S4 Table. Predictors demonstrating statistical significance were subsequently incorporated into adjusted Cox regression models. Three analytical frameworks were applied to explore the relationship between HGI levels and the risk of mortality (Table 3). For 30-day mortality, the HRs were 0.79 (95% CI 0.73–0.87) in the model without adjustments (model 1), 0.8 (95% CI 0.73–0.87) in the model with partial adjustments (model 2), and 0.84(95% CI 0.75–0.93) after full covariate adjustment (model 3), with HGI analyzed as a continuous predictor. For 90-day mortality, the HRs were 0.82 (95% CI: 0.75–0.89), 0.83 (95% CI: 0.76–0.90), and 0.86 (95% CI: 0.79–0.95) for the respective models. Additionally, when HGI was evaluated as a categorical variable divided into tertiles, the HRs for 30-day mortality were: 1.00 (reference) for T1, 0.47(0.36–0.63) for T2, and 0.51 (0.39–0.68) for T3 in model 1. In model 3, these HRs were 1.00 (reference) for T1, 0.6 (0.44–0.81) for T2, and 0.55 (0.41–0.74) for T3. Similar trends were noted for 90-day mortality.

RCS modeling demonstrated a distinct L-shaped association between HGI and the risk of all-cause mortality at both 30 and 90 days (Fig 4). As shown in Table 4, threshold effect analyses identified turning points at 0.692 for 30-day mortality and 0.472 for 90-day mortality (P = 0.002, and P < 0.001, respectively). For 30-day mortality, an HGI < 0.692 corresponded to a 47.7% lower risk of death per unit increase (HR 0.523, 95% CI: 0.419–0.652, P < 0.001). Similarly, for 90-day mortality, an HGI < 0.472 was linked to a 40.4% reduction in mortality risk per unit increase (HR 0.596, 95% CI: 0.486–0.730, P < 0.001). Nonetheless, once HGI surpassed the inflection points, elevated HGI levels did not show a statistically significant correlation with all-cause mortality risk.

**Table 3. Cox proportional hazard ratios (HR) for all-cause mortality.**

| Variable | Model 1 | | Model 2 | | Model 3 | |
|---|---|---|---|---|---|---|
| | HR (95%CI) | P-value | HR (95%CI) | P-value | HR (95%CI) | P-value |
| **30-day mortality** | | | | | | |
| HGI (per unit) | 0.79 (0.73~0.87) | <0.001 | 0.8 (0.73~0.87) | <0.001 | 0.84 (0.75~0.93) | 0.001 |
| HGI (tertiles) | | | | | | |
| T1(HGI<−0.481) | 1(ref) | | 1(ref) | | 1(ref) | |
| T2(−0.481≤HGI<0.066) | 0.47 (0.36~0.63) | <0.001 | 0.45 (0.34~0.6) | <0.001 | 0.6 (0.44~0.81) | 0.001 |
| T3(0.066≤HGI) | 0.51 (0.39~0.68) | <0.001 | 0.49 (0.37~0.65) | <0.001 | 0.55 (0.41~0.74) | <0.001 |
| *P* for trend | | <0.001 | | <0.001 | | <0.001 |
| **90-day mortality** | | | | | | |
| HGI (per unit) | 0.82 (0.75~0.89) | <0.001 | 0.83 (0.76~0.9) | <0.001 | 0.86 (0.79~0.95) | 0.003 |
| HGI (tertiles) | | | | | | |
| T1(HGI<−0.481) | 1(ref) | | 1(ref) | | 1(ref) | |
| T2(−0.481≤HGI<0.066) | 0.56 (0.43~0.73) | <0.001 | 0.53 (0.41~0.69) | <0.001 | 0.7 (0.54~0.92) | 0.009 |
| T3(0.066≤HGI) | 0.58 (0.45~0.76) | <0.001 | 0.56 (0.43~0.72) | <0.001 | 0.62 (0.47~0.81) | 0.001 |
| *P* for trend | | <0.001 | | <0.001 | | <0.001 |

Model 1: crude.

Model 2: adjusted for age, gender, race, respiratory rate, SpO2.

Model 3: adjusted for Model 2 plus SOFA, GCS, congestive heart failure, myocardial Infarction, sepsis, WBC, hemoglobin, platelet, BUN, creatinine, potassium, mannitol, diuretic, insulin, vasoactive drug, ventilation, cerebral surgery.

Abbreviations: HGI, hemoglobin glycation index; SpO2, oxygen saturation; SOFA, sequential organ failure assessment; GCS, Glasgow coma scale; WBC, white blood cell; BUN, blood urea nitrogen.

## Subgroup analyses

Additionally, we conducted stratified analyses across multiple clinical variables, including age, gender, sepsis, diabetes, PVD, and myocardial infarction (Figs 5 and 6). Regarding all-cause mortality at both 30 and 90 days, the interaction effects were generally non-significant, with the exception of diabetes (P for interaction = 0.04 and 0.037, respectively).

## Discussion

This study provides novel insights into the association between the HGI and short-term prognosis in individuals diagnosed with ICH. Our results indicate that a lower HGI independently correlates with higher 30-day and 90-day all-cause mortality, offering a valuable predictive marker for clinical outcomes in this patient cohort. Additionally, individuals with the lowest HGI values experienced elevated in-hospital death rates, increased ICU mortality, and prolonged ICU stays. The identification of inflection points for HGI at 0.692 and 0.472 for 30-day and 90-day mortality, respectively, is an innovative aspect of this research, suggesting that HGI could serve as a threshold marker to refine prognostic risk stratification.

Notably, during the preparation of our manuscript, a concurrent study by Sun et al. [12] reported the first association between HGI and 30-day mortality in ICH patients using the same MIMIC-IV database, identifying a threshold effect at HGI of 0.78. Our study extends these preliminary findings in several important ways. First, we examined both 30-day and 90-day mortality outcomes, revealing that the prognostic threshold exhibits time-dependent variation—the difference between the 30-day inflection point (0.692) and 90-day inflection point (0.472) suggests that the prognostic significance of HGI may evolve dynamically beyond the acute phase of illness. Second, our subgroup analyses demonstrated a significant effect modification by diabetes status (P for interaction < 0.05), indicating that the predictive value of HGI is more pronounced in patients with impaired glucose regulation—a finding that adds a new dimension to the biological interpretation of HGI. Third, although our regression equation for calculating predicted HbA1c ($0.013 \times FPG + 4.354$) differed slightly

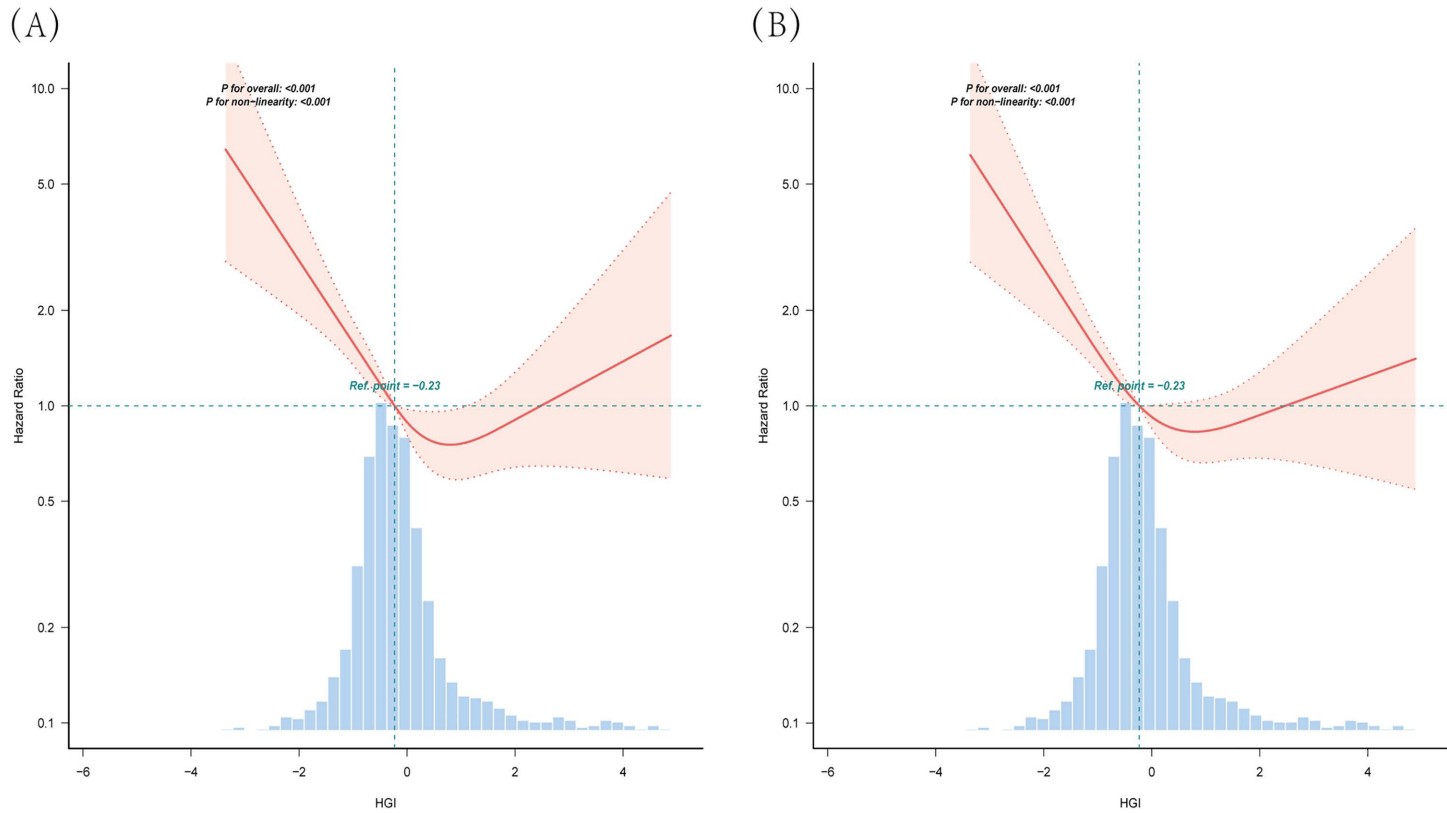

**Fig 4. RCS curves for HGI and hazard ratios.** (A) RCS curve for 30-day mortality. (B) **RCS curve for 90-day mortality.** The median value of HGI serves as the reference. Models are adjusted for all covariates in Model 3 (Table 3). RCS, restricted cubic spline; HGI, hemoglobin glycation index.

**Table 4. The nonlinear relationship between HGI and all-cause mortality.**

| Models | HR (95%CI) | P-value |
|---|---|---|
| **30-day mortality** | | |
| Infection point | 0.692 | |
| Fitting model by two-piecewise linear regression | | |
| <0.692 | 0.523 (0.419,0.652) | < 0.001 |
| ≥0.692 | 1.115 (0.755,1.647) | 0.5847 |
| Likelihood Ratio test | | 0.002 |
| **90-day mortality** | | |
| Infection point | 0.472 | |
| Fitting model by two-piecewise linear regression | | |
| <0.472 | 0.596 (0.486,0.73) | < 0.001 |
| ≥0.472 | 1.242 (0.939,1.642) | 0.1284 |
| Likelihood Ratio test | | <0.001 |

from that of Sun et al. ($0.012 \times glucose + 4.383$), both studies consistently confirmed the association between low HGI and adverse outcomes, reinforcing the robustness of this metric. Together, these complementary findings suggest that HGI may serve as a valuable tool for metabolic risk assessment in ICH patients, and future research should further elucidate its predictive value across different clinical subgroups and time points.

| Subgroup | Group | Count | HR(95%CI) | | P for interaction |
|---|---|---|---|---|---|
| Age | | | | | 0.714 |
| <65 | | | | | |
| | T1 | 156 | 1(Ref) | | |
| | T2 | 141 | 0.56 (0.27~1.17) | | |
| | T3 | 136 | 0.59 (0.28~1.23) | | |
| >=65 | | | | | |
| | T1 | 278 | 1(Ref) | | |
| | T2 | 308 | 0.63 (0.45~0.87) | | |
| | T3 | 299 | 0.57 (0.41~0.79) | | |
| Gender | | | | | 0.254 |
| Female | | | | | |
| | T1 | 225 | 1(Ref) | | |
| | T2 | 205 | 0.54 (0.35~0.82) | | |
| | T3 | 186 | 0.7 (0.47~1.03) | | |
| Male | | | | | |
| | T1 | 209 | 1(Ref) | | |
| | T2 | 244 | 0.82 (0.52~1.28) | | |
| | T3 | 249 | 0.55 (0.34~0.89) | | |
| Sepsis | | | | | 0.151 |
| No | | | | | |
| | T1 | 270 | 1(Ref) | | |
| | T2 | 304 | 0.51 (0.33~0.8) | | |
| | T3 | 292 | 0.52 (0.34~0.81) | | |
| Yes | | | | | |
| | T1 | 164 | 1(Ref) | | |
| | T2 | 145 | 0.7 (0.46~1.07) | | |
| | T3 | 143 | 0.59 (0.39~0.89) | | |
| Diabetes | | | | | 0.04 |
| No | | | | | |
| | T1 | 361 | 1(Ref) | | |
| | T2 | 373 | 0.71 (0.5~1.01) | | |
| | T3 | 177 | 0.81 (0.52~1.25) | | |
| Yes | | | | | |
| | T1 | 73 | 1(Ref) | | |
| | T2 | 76 | 0.45 (0.23~0.89) | | |
| | T3 | 258 | 0.37 (0.22~0.63) | | |
| PVD | | | | | 0.118 |
| No | | | | | |
| | T1 | 416 | 1(Ref) | | |
| | T2 | 419 | 0.68 (0.5~0.92) | | |
| | T3 | 397 | 0.61 (0.45~0.83) | | |
| Yes | | | | | |
| | T1 | 18 | 1(Ref) | | |
| | T2 | 30 | 0.03 (0~0.32) | | |
| | T3 | 38 | 0.13 (0.02~0.78) | | |
| MI | | | | | 0.622 |
| No | | | | | |
| | T1 | 399 | 1(Ref) | | |
| | T2 | 418 | 0.68 (0.5~0.94) | | |
| | T3 | 380 | 0.59 (0.43~0.82) | | |
| Yes | | | | | |
| | T1 | 35 | 1(Ref) | | |
| | T2 | 31 | 0.22 (0.07~0.67) | | |
| | T3 | 55 | 0.27 (0.1~0.72) | | |

0    0.5    1    1.5

**Fig 5. Forest plots for subgroup analysis of HGI and 30-day all-cause mortality, adjusted for covariates in Model 3 (Table 3), except the stratification variable.** HGI, hemoglobin glycation index; PVD, peripheral vascular disease; MI, myocardial infarction.

| Subgroup | Group | Count | HR(95%CI) | P for interaction |
|---|---|---|---|---|
| Age | | | | 0.705 |
| <65 | | | | |
| | T1 | 156 | 1(Ref) | |
| | T2 | 141 | 0.74 (0.38~1.43) | |
| | T3 | 136 | 0.73 (0.37~1.43) | |
| >=65 | | | | |
| | T1 | 278 | 1(Ref) | |
| | T2 | 308 | 0.72 (0.54~0.97) | |
| | T3 | 299 | 0.63 (0.47~0.85) | |
| Gender | | | | 0.211 |
| Female | | | | |
| | T1 | 225 | 1(Ref) | |
| | T2 | 205 | 0.72 (0.5~1.04) | |
| | T3 | 186 | 0.79 (0.55~1.13) | |
| Male | | | | |
| | T1 | 209 | 1(Ref) | |
| | T2 | 244 | 0.83 (0.56~1.25) | |
| | T3 | 249 | 0.62 (0.41~0.94) | |
| Sepsis | | | | 0.414 |
| No | | | | |
| | T1 | 270 | 1(Ref) | |
| | T2 | 304 | 0.62 (0.42~0.92) | |
| | T3 | 292 | 0.64 (0.43~0.95) | |
| Yes | | | | |
| | T1 | 164 | 1(Ref) | |
| | T2 | 145 | 0.83 (0.57~1.21) | |
| | T3 | 143 | 0.62 (0.42~0.91) | |
| Diabetes | | | | 0.037 |
| No | | | | |
| | T1 | 361 | 1(Ref) | |
| | T2 | 373 | 0.85 (0.63~1.16) | |
| | T3 | 177 | 0.91 (0.62~1.34) | |
| Yes | | | | |
| | T1 | 73 | 1(Ref) | |
| | T2 | 76 | 0.54 (0.29~0.99) | |
| | T3 | 258 | 0.42 (0.26~0.69) | |
| PVD | | | | 0.086 |
| No | | | | |
| | T1 | 416 | 1(Ref) | |
| | T2 | 419 | 0.79 (0.6~1.04) | |
| | T3 | 397 | 0.7 (0.53~0.92) | |
| Yes | | | | |
| | T1 | 18 | 1(Ref) | |
| | T2 | 30 | 0.12 (0.03~0.56) | |
| | T3 | 38 | 0.1 (0.02~0.52) | |
| MI | | | | 0.643 |
| No | | | | |
| | T1 | 399 | 1(Ref) | |
| | T2 | 418 | 0.79 (0.59~1.06) | |
| | T3 | 380 | 0.66 (0.49~0.88) | |
| Yes | | | | |
| | T1 | 35 | 1(Ref) | |
| | T2 | 31 | 0.22 (0.08~0.58) | |
| | T3 | 55 | 0.29 (0.12~0.67) | |

**Fig 6. Forest plots for subgroup analysis of HGI and 90-day all-cause mortality, adjusted for covariates in Model 3 (Table 3), except the stratification variable.** HGI, hemoglobin glycation index; PVD, peripheral vascular disease; MI, myocardial infarction.

Diabetes mellitus is widely acknowledged as a major contributor to cerebrovascular diseases, including ICH [15]. Chronic hyperglycemia contributes to impaired endothelial function and vascular damage, thereby elevating the risk of hemorrhagic events. Microvascular sequelae of diabetes, including retinopathy and nephropathy, show strong correlations with cerebrovascular pathologies. In a one-year prospective study involving 100 individuals diagnosed with ICH, Tseng WC et al. reported that diabetes mellitus was independently associated with poor functional outcomes in ICH, even after adjusting for potential confounders [16].

HbA1c is a commonly utilized indicator representing long-term glycemic levels across the prior two to three months, and plays a pivotal role in identifying and monitoring diabetes mellitus [17]. Evidence suggests that higher HbA1c levels correlate with a greater incidence and severity of ICH [15,18]. Liu H et al. demonstrated that, among individuals suffering from ICH and no prior record of diabetes, an elevated HbA1c level (≥6.5%) correlated with a significantly increased risk of unfavorable neurological prognoses and higher death rates [19]. However, while HbA1c offers valuable insight into long-term blood glucose management, it has inherent limitations. It does not account for glycemic variability or transient hyperglycemic spikes, which have been increasingly recognized as critical contributors to vascular injury. Moreover, non-glycemic influences such as hemoglobinopathies, anemia, and variations in red blood cell lifespan can affect HbA1c values, potentially leading to misinterpretation of glycemic control.

To address these limitations, HGI has been introduced as a complementary metric. By quantifying the discrepancy between an individual's glycation propensity and ambient glucose concentration, HGI provides unique insights into the biological mechanisms underlying glycemic exposure and vascular risk. An increasing number of studies suggest that HGI correlates with adverse clinical outcomes across multiple disease states, including cardiovascular disease, chronic kidney disease, diabetes, and cerebrovascular disorders. In a large cohort study involving over 11,000 individuals diagnosed with both type 2 diabetes and coronary artery disease, Lin et al. identified a nonlinear relationship involving HGI and cardio-vascular outcomes, with extreme HGI values significantly increasing the likelihood of serious cardiovascular complications [20]. Similarly, a retrospective analysis of patient data leveraging the MIMIC-IV clinical dataset reported that reduced HGI values were independently linked to elevated short-, intermediate-, and long-term mortality among severely ill individuals suffering from myocardial infarction [10]. In the context of renal disease, Lin et al. found that among individuals diagnosed type 2 diabetes and initially low chronic kidney disease risk, increased HGI levels predicted a more rapid decline in renal function [21]. In diabetes management, a higher hemoglobin glycation index has also been associated with greater risks of microvascular complications and macrovascular conditions, including diabetic retinopathy, nephropathy, and cardio-vascular events [11]. Moreover, Huang et al. recently demonstrated a nonlinear association between HGI and death risk among individuals diagnosed with ischemic stroke, wherein reduced HGI showed a significant correlation with heightened short-term fatality, highlighting HGI's prognostic relevance in acute cerebrovascular conditions [22].

Thus, investigating the relationship between HGI and ICH may offer deeper insights into how glycemic patterns influence cerebrovascular pathology, beyond what is reflected by HbA1c alone. In the present analysis, we identified a significant association between reduced HGI and elevated short-term mortality risk among individuals with ICH. These findings may be attributed to the following potential explanations. First, acute stress responses in severe ICH can trigger sympathetic and hypothalamic–pituitary–adrenal axis activation, elevating catecholamines and cortisol, which promote transient hyperglycemia [23,24]. This acute glucose elevation triggered by physiological stress may substantially increase the estimated HbA1c levels derived from plasma glucose concentrations, leading to a relative decrease in HGI. Consequently, a low HGI in the acute setting may not solely reflect an individual's intrinsic propensity for protein glycation but could also serve as a surrogate marker for the severity of the physiological stress response. Although our multivariable models were adjusted for several indicators of clinical severity, including SOFA score, GCS, and the use of vasoactive drugs, to mitigate this confounding effect, residual confounding cannot be entirely excluded. Li et al. carried out a multicenter investigation that enrolled 586 individuals diagnosed with spontaneous ICH and determined that stress-induced hyperglycemia closely correlated with adverse early clinical outcomes [25]. Similarly, a retrospective analysis by Zhang et al. including 880 ICH

patients showed that even after adjusting for confounders, the stress hyperglycemia ratio remained a stand-alone prognostic indicator of unfavorable outcomes, underscoring its strong prognostic value [26]. These findings imply that acute-phase stress hyperglycemia can distort the assessment of chronic glycemic control, resulting in an artificially low HGI that correlates with worse clinical prognoses among patients with ICH. The more pronounced association observed in patients with diabetes suggests that HGI may capture different risk information depending on the metabolic context. This finding supports the notion that in non-diabetic patients, HGI may be more susceptible to acute stress-related fluctuations, whereas in diabetic patients, it may better reflect intrinsic glycation tendencies with independent prognostic value. Future studies incorporating serial glucose measurements or direct biomarkers of the stress response (e.g., cortisol, catecholamines) would help to further disentangle the independent contribution of HGI from that of acute stress hyperglycemia in ICH prognosis. Second, anemia might contribute to this effect by lowering HbA1c values due to reduced erythrocyte lifespan, resulting in underestimation of chronic glycemia and thus lower HGI. This occurs because the measured HbA1c, the "observed" value in the HGI calculation, is artifactually decreased, while the HbA1c predicted from plasma glucose remains unchanged, leading to a more negative HGI. In our cohort, hemoglobin concentrations were notably reduced among individuals with low HGI (12.2 g/dL vs. 12.7 g/dL, P < 0.001). Anemia has been correlated with a heightened probability of hematoma expansion, with every 1 g/dL reduction in hemoglobin tied to roughly a 10% elevation in the chance of such progression [27]. It is also correlated with larger hematoma volumes and more severe neurological deficits [28]. Meta-analytic evidence indicates that anemia significantly elevates short- and long-term death rates among individuals with ICH [29]. Collectively, this body of evidence implies that anemia not merely has a direct negative impact on patient prognosis, but may also artifactually lower HGI, thereby contributing to the observed link between reduced HGI and heightened risk of fatality.

Our RCS modeling additionally demonstrated a clear L-shaped pattern linking HGI to all-cause mortality at both 30 and 90 days, highlighting the nonlinear nature of this association. Breakpoint analysis pinpointed critical values at HGI levels of 0.692 and 0.472 corresponding to 30-day and 90-day mortality, respectively. Below these thresholds, each incremental rise in HGI correlated with substantial reductions in mortality likelihood—47.7% for 30-day and 40.4% for 90-day mortality. However, once these inflection points were exceeded, further increases in HGI did not show statistically meaningful correlations with shifts in mortality probability. Notably, the difference between the 30-day and 90-day thresholds (0.692 vs. 0.472) suggests that the prognostic significance of HGI may evolve over time. This time-dependent threshold effect, which has not been previously reported, provides a rationale for dynamic monitoring of HGI beyond the initial presentation. These results highlight that low HGI may reflect adverse physiological states such as stress hyperglycemia or anemia, which are strongly linked to early mortality, whereas higher HGI levels may have diminishing prognostic utility beyond a certain point. Thus, HGI should be interpreted as a dynamic, context-dependent biomarker, and its prognostic value may be most meaningful in individuals exhibiting relatively low HGI levels.

Subgroup analyses further explored whether the association between HGI and mortality was modified by baseline clinical characteristics. We observed consistent protective effects of higher HGI (≥ 0.066) on short-term overall mortality across most subgroups. Nevertheless, a statistically notable effect modification was identified in relation to diabetes status, suggesting that the prognostic relevance of HGI may vary depending on whether diabetes is present. Specifically, the inverse relationship between HGI and mortality was not evident among patients without diabetes, indicating that the predictive utility of HGI may be more pronounced in those with impaired glucose metabolism. This finding, which was not reported in the concurrent study by Sun et al. [12], adds granularity to the clinical application of HGI and suggests that diabetes status should be considered when interpreting HGI for risk stratification. This finding is biologically plausible, as HGI reflects the difference between measured HbA1c and the level predicted from blood glucose concentrations, a phenomenon likely to be more relevant in the context of abnormal glycemic regulation.

In addition to the subgroup findings, we carefully evaluated the potential impact of missing data on our cohort composition. The observed differences between patients with missing HbA1c measurements and the included cohort are

consistent with real-world clinical testing practices and do not indicate systematic selection bias. The higher prevalence of diabetes and insulin use in the included group reflects routine HbA1c monitoring in patients with known glucose metabolism disorders, whereas older age in this group aligns with the tendency to perform comprehensive metabolic assessments in elderly patients with higher comorbidity burden. Conversely, the excluded group had slightly higher SOFA scores and more frequent use of mannitol, vasoactive drugs, and cerebral surgery, indicating greater acute illness severity. In such critically ill patients, clinicians prioritize life-saving interventions over non-urgent tests like HbA1c, explaining the missing data. The higher mortality in the excluded group is a logical consequence of their greater baseline severity, consistent with the well-established relationship between multi-organ dysfunction and poor outcomes. Importantly, the two groups were well balanced in core ICH prognostic factors (e.g., GCS score, mechanical ventilation use), supporting that the missing HbA1c data are driven by clinical priorities rather than selection bias. Therefore, the included cohort remains representative of the ICH population in the MIMIC-IV database, and our main findings are robust.

Overall, our findings provide novel insights into the prognostic value of HGI among individuals suffering from ICH. The results suggest that particularly reduced HGI values could act as an early warning indicator for deterioration in severely ill ICH patients, warranting closer clinical monitoring and possibly more aggressive management. However, there has several limitations that should be acknowledged. First, HGI was estimated through a regression-based method utilizing admission FBG and HbA1c levels, which were measured at a single measurement point. This approach may insufficiently capture long-term glycemic trends or the dynamic nature of glucose fluctuations during hospitalization. Second, important clinical variables such as hematoma volume, location, and expansion—which are known to be strong predictors of ICH outcomes—were not available in the MIMIC database and therefore could not be included in the analysis. This represents a significant limitation, as these radiographic factors are major determinants of neurological deterioration and mortality in ICH patients [30]. If these factors are correlated with HGI, the inability to adjust for them could introduce residual confounding. For example, patients with larger hematomas may experience more profound stress responses, leading to lower HGI values. Such confounding might lead to an overestimation of the independent association between HGI and mortality. To partially mitigate this concern, we included the GCS score in our fully adjusted models. GCS is a well-established clinical surrogate for the neurological impact of ICH and has been shown to correlate strongly with hematoma volume and location [31,32], this adjustment is imperfect. Ideally, future prospective cohort studies should be designed to collect standardized neuroimaging data to rigorously evaluate whether HGI provides incremental prognostic value beyond established ICH-specific radiographic predictors. Lastly, due to the study's retrospective nature and dependence on data from a single institution, inherent selection bias and residual confounding cannot be fully excluded, even with multivariable adjustments, and replication in diverse, multi-institutional populations is strongly encouraged.

## Conclusion

In summary, our study demonstrates that HGI exhibited a significant relationship with heightened short-term overall mortality among individuals diagnosed with ICH. The observed nonlinear L-shaped relationship, in which the highest risk is concentrated at lower HGI levels, highlights the nuanced role of glycemic markers in acute cerebrovascular pathology. These findings suggest that HGI could be utilized as a novel biomarker to aid early risk stratification in ICH, complementing traditional prognostic factors. Incorporating HGI into clinical assessment may help identify high-risk patients who would benefit from intensified monitoring or therapeutic interventions. Future prospective studies are warranted to validate the prognostic value of HGI.

## Supporting information

**S1 Table. missing number of variables.**
(DOCX)

**S2 Table. Baseline characteristics between patients with missing HbA1c measurements and included ICH patients.**
(DOCX)

**S3 Table. Baseline characteristics between 30-day survival and 30-day mortality group.**
(DOCX)

**S4 Table. Univariate cox proportional analysis for 30-day mortality.**
(DOCX)

## Acknowledgments

We acknowledge the MIMIC-IV team for their ongoing efforts in maintaining and improving the dataset.

## Author contributions

**Data curation:** Jia Xu.

**Formal analysis:** Jia Xu.

**Supervision:** Yongfeng Ni.

**Validation:** Guangdong Wang, Yongfeng Ni.

**Writing – original draft:** Jia Xu.

**Writing – review & editing:** Guangdong Wang, Yongfeng Ni.

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
