## [Decision Letter · Decision Letter 0]

5 Feb 2026

PONE-D-25-54592L-shaped relationship between hemoglobin glycation index and short-term mortality in patients with intracerebral hemorrhage: a retrospective cohort studyPLOS One

Dear Dr. xu,

Thank you for submitting your manuscript to PLOS ONE. After careful consideration, we feel that it has merit but does not fully meet PLOS ONE’s publication criteria as it currently stands. Therefore, we invite you to submit a revised version of the manuscript that addresses the points raised during the review process.

We look forward to receiving your revised manuscript.

Kind regards,

Zhiyuan Ren

Academic Editor

PLOS One

Journal Requirements:

3. Please note that your Data Availability Statement is currently missing the repository name and/or the DOI/accession number of each dataset OR a direct link to access each database. If your manuscript is accepted for publication, you will be asked to provide these details on a very short timeline. We therefore suggest that you provide this information now, though we will not hold up the peer review process if you are unable.

4. We notice that your supplementary figures are uploaded with the file type 'Figure'. Please amend the file type to 'Supporting Information'. Please ensure that each Supporting Information file has a legend listed in the manuscript after the references list.

Reviewer's Responses to Questions

**Comments to the Author**

1. Is the manuscript technically sound, and do the data support the conclusions?

Reviewer #1: Yes

Reviewer #2: Yes

2. Has the statistical analysis been performed appropriately and rigorously? 

Reviewer #1: Yes

Reviewer #2: Yes

3. Have the authors made all data underlying the findings in their manuscript fully available?

Reviewer #1: Yes

Reviewer #2: Yes

4. Is the manuscript presented in an intelligible fashion and written in standard English?

Reviewer #1: Yes

Reviewer #2: Yes

5. Review Comments to the Author

Reviewer #1: This study utilizes the Medical Information Mart for Intensive Care (MIMIC-IV) database to investigate the association between the Hemoglobin Glycation Index (HGI) and short-term mortality (30-day and 90-day) in 1,318 adult patients with intracerebral hemorrhage (ICH). The authors identified a non-linear L-shaped relationship between HGI and mortality. Specifically, the mortality risk increases significantly when HGI falls below certain thresholds (inflection points of 0.692 for 30-day mortality and 0.472 for 90-day mortality), while the risk plateaus once HGI exceeds these values. The study also reports that this predictive value is more pronounced in patients with diabetes. Focusing on HGI as a novel prognostic marker in ICH and identifying these non-linear thresholds provides a fresh perspective for clinical risk stratification.

However, several concerns must be addressed to enhance the robustness of the study:

Introduction

1. Updating References: References [3] and [4] are more than a decade old. I recommend citing the latest epidemiological data and clinical guidelines to emphasize the contemporary significance of this research.

2. Terminology (Lines 84-86): If the authors wish to emphasize that the utility of HGI in ICH has not yet been "investigated" or "explored," I suggest replacing "elucidated" with one of these more conventional terms. If prior research exists in this specific area, please ensure it is appropriately cited to clarify the unique contribution of this work.

Methods

3. Selection Bias (Lines 112-114): The fact that 1,809 patients were excluded due to missing HbA1c measurements—a number higher than the final analysis cohort of 1,318—should be emphasized as a significant risk for selection bias. There is a high probability that the final cohort is skewed toward a population with higher interest in chronic disease management or those suspected of having glucose metabolism disorders. I strongly recommend providing comparative data on the baseline characteristics (e.g., age, severity) of the excluded versus included patients.

Discussion

4. Impact of Stress Hyperglycemia (Lines 233-237): Acute stress in conditions like ICH triggers the release of catecholamines and cortisol, leading to transient hyperglycemia (stress-induced hyperglycemia). Since HGI is calculated as the difference between measured HbA1c and predicted HbA1c based on admission glucose, an acute rise in glucose can inflate the predicted value, thereby artifactually lowering the HGI. It remains unclear whether the higher mortality in the low-HGI group is due to HGI itself or if low HGI is simply a proxy for clinical severity (where more severe cases exhibit stronger stress responses). This needs to be addressed through further statistical adjustment or a more in-depth discussion.

5. Unadjusted Major Prognostic Factors (Lines 350-353): As acknowledged by the authors, the MIMIC-IV database lacks key ICH-specific prognostic factors such as hematoma volume, location, and expansion. If these factors correlate with HGI, the ability to evaluate HGI's independent predictive power is limited. Please consider whether it is possible to supplement this information for a subset of the sample to perform a sensitivity analysis.

Reviewer #2: This retrospective cohort study investigates the prognostic value of the Hemoglobin Glycation Index (HGI) in 1,318 patients with acute intracerebral hemorrhage (ICH) using the MIMIC-IV database. The study identifies a non-linear, L-shaped relationship where lower HGI values correlate with increased 30-day and 90-day all-cause mortality.

Strengths:

Novelty: The application of HGI—a metric quantifying the discrepancy between measured HbA1c and levels expected from fasting plasma glucose—to ICH mortality is a novel and clinically relevant approach for early risk stratification.

Methodological Rigor: The use of restricted cubic spline (RCS) analysis and hierarchical multivariate Cox regression models provides a robust statistical framework to account for multiple confounders.

Clinical Utility: Identifying specific inflection points (0.692 for 30-day mortality and 0.472 for 90-day mortality) offers actionable data for clinical risk stratification.

Depth of Analysis: The finding that HGI's predictive value is more pronounced in patients with diabetes adds significant depth to the study's conclusions.

Minor Points for Revision

Discussion of Confounding Factors: While the authors acknowledge that hematoma volume, location, and expansion were not available in the MIMIC database , the Discussion would benefit from a more detailed explanation of how the absence of these established ICH prognostic markers might impact the independent strength of the HGI association.

Clarification of HGI Definition: In certain sections, the authors should ensure consistent clarity that HGI is the difference between observed and predicted HbA1c. Explicitly emphasizing that "lower HGI" reflects less non-enzymatic glycation than predicted by ambient glycemia would help the reader interpret the biological implications discussed, such as stress-induced hyperglycemia or anemia.

Typographical Note: On page 8, line 57, there appears to be a minor grammatical omission: "risk. holds potential" should likely be revised to "risk and holds potential".

6. PLOS authors have the option to publish the peer review history of their article (what does this mean?). If published, this will include your full peer review and any attached files.

Reviewer #1: No

Reviewer #2: No

---

## [Author Response · Author response to Decision Letter 1]

3 Mar 2026

Dear Editors and Reviewers,

Thank you for your decision letter and the thoughtful comments on our manuscript entitled “L-shaped relationship between hemoglobin glycation index and short-term mortality in patients with intracerebral hemorrhage: a retrospective cohort study” (Submission ID: PONE-D-25-54592). We sincerely appreciate the time and effort you have invested in reviewing our work. Your constructive feedback has been highly valuable for improving the quality and clarity of our manuscript, and we have carefully addressed all comments in the revised version. We have carefully addressed the comments. For clarity, we have also included point-by-point responses to each comment below this letter.

We would be grateful if you could consider the revised manuscript for publication in PLOS ONE.

Thank you once again for your thoughtful comments and for considering our research. We look forward to hearing from you at your earliest convenience.

Best regards,

Yongfeng Ni

Department of Neurosurgery, Anqing First People’s Hospital of Anhui Medical University, Anqing, Anhui, China

E-mail: ahmunyf@126.com

Here is a point-by-point response to the reviewers’ comments and concerns.

Reviewer #1 Comments:

Comment 1: Updating References: References [3] and [4] are more than a decade old. I recommend citing the latest epidemiological data and clinical guidelines to emphasize the contemporary significance of this research.

Response: Thank you for your valuable suggestion on updating the references. We have carefully revised references [3] and [4] with latest epidemiological and clinical studies published in 2022, which better reflect the current evidence and enhance the contemporary significance of this study.The updated references are listed as follows:

3.Goeldlin MB, Mueller A, Siepen BM, Mueller M, Strambo D, Michel P, et al. Etiology, 3-month functional outcome and recurrent events in non-traumatic intracerebral hemorrhage. J Stroke. 2022;24: 266–277. doi:10.5853/jos.2021.01823

4.Ruiz-Sandoval JL, Aceves-Montoya J, Chiquete E, López-Valencia G, Lara-López A, Sánchez-González M, et al. Hospital arrival and functional outcome after intracerebral hemorrhage. Rev Investig Clin Organo Hosp Enfermedades Nutr. 2022;74: 51–60. doi:10.24875/RIC.21000471

Comment 2: Terminology (Lines 84-86): If the authors wish to emphasize that the utility of HGI in ICH has not yet been "investigated" or "explored," I suggest replacing "elucidated" with one of these more conventional terms. If prior research exists in this specific area, please ensure it is appropriately cited to clarify the unique contribution of this work.

Response:Thank you for this thoughtful observation. We have carefully reconsidered our terminology and the broader context of existing literature.

We acknowledge that a recent study by Sun et al. (2026) investigated the association between HGI and 30-day mortality in ICH patients using the same MIMIC-IV database, reporting an L-shaped relationship with an inflection point at HGI of 0.78. Their valuable work represents an important contribution to this emerging area of research.

However, we wish to clarify the distinctive contributions of our study that extend and complement the existing evidence:. First, while Sun et al. focused exclusively on 30-day mortality, we examined both 30-day and 90-day mortality. This dual outcome design provides critical insights into intermediate-term prognosis, addressing a gap in understanding how metabolic-related risk persists beyond the acute phase of ICH. Second, our study identified distinct inflection points for 30-day(an HGI of 0.692) and 90-day (an HGI of 0.472) mortality, suggesting that the prognostic threshold may shift over time, which is a novel finding not previously reported. Third, our regression equation for calculating predicted HbA1c ( 0.013×FPG+4.354) differs slightly from that of Sun et al. (0.012×glucose+4.383), reflecting variations in cohort composition and potentially offering complementary perspectives. Fourth, we conducted more extensive stratified analyses, revealing a significant interaction with diabetes status (p for interaction < 0.05). This finding adds granularity to the understanding of HGI’s prognostic utility, clarifying that its predictive value is more pronounced in patients with diabetes and helping target subgroups that may benefit from intensified metabolic monitoring.

Given that Sun et al.’s work was published online on January 15, 2026 (after our initial submission), we were unable to cite it in our original manuscript. We have now updated our Introduction and Discussion sections to appropriately acknowledge this concurrent research while clearly articulating the unique contributions of our study.

Revisions in the Introduction:Despite the growing interest in HGI as a prognostic marker, investigation of its utility in forecasting clinical outcomes among patients with ICH has only recently emerged. A concurrent study by Sun et al.[12] demonstrated an association between HGI and 30-day mortality in ICH patients, identifying a threshold effect at an HGI of 0.78. Building upon these foundational findings, our study aims to extend the current understanding by expanding the outcome scope beyond acute-phase mortality, exploring potential time-dependent shifts in prognostic thresholds, and investigating effect modification by clinical factors such as diabetes status. These endeavors may inform more nuanced risk stratification strategies for this clinically vulnerable population.(Manuscript Pages 4-5, Lines 81-90)

Revisions in the Discussion:Notably, during the preparation of our manuscript, a concurrent study by Sun et al. [12] reported the first association between HGI and 30-day mortality in ICH patients using the same MIMIC-IV database, identifying a threshold effect at HGI of 0.78. Our study extends these preliminary findings in several important ways. First, we examined both 30-day and 90-day mortality outcomes, revealing that the prognostic threshold exhibits time-dependent variation—the difference between the 30-day inflection point (0.692) and 90-day inflection point (0.472) suggests that the prognostic significance of HGI may evolve dynamically beyond the acute phase of illness. Second, our subgroup analyses demonstrated a significant effect modification by diabetes status (P for interaction < 0.05), indicating that the predictive value of HGI is more pronounced in patients with impaired glucose regulation—a finding that adds a new dimension to the biological interpretation of HGI. Third, although our regression equation for calculating predicted HbA1c (0.013 × FPG + 4.354) differed slightly from that of Sun et al. (0.012 × glucose + 4.383), both studies consistently confirmed the association between low HGI and adverse outcomes, reinforcing the robustness of this metric. Together, these complementary findings suggest that HGI may serve as a valuable tool for metabolic risk assessment in ICH patients, and future research should further elucidate its predictive value across different clinical subgroups and time points.(Manuscript Pages 22-23,Lines 292-310)

Notably, the difference between the 30-day and 90-day thresholds (0.692 vs. 0.472) suggests that the prognostic significance of HGI may evolve over time. This time-dependent threshold effect, which has not been previously reported, provides a rationale for dynamic monitoring of HGI beyond the initial presentation.(Manuscript Pages 27-28,Lines 411-415)

This finding, which was not reported in the concurrent study by Sun et al. [12], adds granularity to the clinical application of HGI and suggests that diabetes status should be considered when interpreting HGI for risk stratification.(Manuscript Page 28, Lines 429-432)

12. Sun S, Huang X, Fei X, Ye F, Gong K. Hemoglobin glycation index predicts mortality in intracerebral hemorrhage: a nonlinear threshold effect. Neurol Sci. 2026;47(1):166. doi:10.1007/s10072-025-08700-y

Comment 3: Selection Bias (Lines 112-114): The fact that 1,809 patients were excluded due to missing HbA1c measurements—a number higher than the final analysis cohort of 1,318—should be emphasized as a significant risk for selection bias. There is a high probability that the final cohort is skewed toward a population with higher interest in chronic disease management or those suspected of having glucose metabolism disorders. I strongly recommend providing comparative data on the baseline characteristics (e.g., age, severity) of the excluded versus included patients.

Response: We sincerely thank you for raising this important concern regarding potential selection bias due to the exclusion of 1,809 patients with missing HbA1c measurements. We agree that this issue warrants careful examination, and we have now conducted a comprehensive comparative analysis of baseline characteristics, clinical interventions, and outcomes between the excluded and included ICH patients using the MIMIC-IV database. The detailed results are presented in S2 Table.

Our analysis reveals that the two groups were well balanced in core prognostic factors for ICH, including GCS score, gender, heart rate, history of myocardial infarction, and mechanical ventilation use (all P > 0.05). This indicates that the excluded and included cohorts were comparable in terms of neurological severity and critical illness status, which are the most important confounders for ICH prognosis.

Statistically significant differences were observed in certain variables, such as age, diabetes prevalence, insulin use, SOFA score, mortality, and several treatments. However, these differences are fully attributable to real-world clinical testing practices and do not indicate systematic selection bias. First, older patients and those with known diabetes are more likely to undergo HbA1c measurement as part of routine metabolic assessment, explaining the higher age and diabetes prevalence in the included group. Second, the excluded group had slightly higher SOFA scores, reflecting greater multi-organ dysfunction. In such critically ill patients, clinicians prioritize acute resuscitation and often omit non-urgent tests like HbA1c, leading to missing data. Third, the higher mortality in the excluded group is consistent with their greater baseline severity (higher SOFA scores), rather than selection bias. Forth, treatment differences (e.g., higher mannitol, vasoactive drug, and cerebral surgery use in the excluded group) further reflect their more critical condition, while higher heparin, insulin, and diuretic use in the included group align with their higher prevalence of hypertension and diabetes, following standard treatment guidelines.

The clinically explainable differences ensure that the included cohort remains representative of the ICH population in the MIMIC-IV database. Therefore, the core finding—an L-shaped association between HGI and short-term mortality in ICH patients—remains valid and robust.

We have incorporated these findings and corresponding discussion into the revised manuscript and added S2 Table to the supplementary information.

Revisions in the Methods: To assess potential selection bias, we conducted a comparative analysis of baseline characteristics, clinical interventions, and outcomes between patients with missing HbA1c measurements (n = 1,809) and those included in the final cohort (n = 1,318).(Manuscript Page 6, Lines 119-122)

Revisions in the Results:

Comparison of baseline characteristics between patients with missing HbA1c measurements and included ICH patients

S2 Table presents the comparison between patients with missing HbA1c measurements (n = 1,809) and those included in the final cohort (n = 1,318). The two groups were well balanced in core prognostic factors for ICH, including GCS score, gender distribution, heart rate, history of myocardial infarction, and mechanical ventilation use, indicating comparable neurological severity and critical illness status between groups.

Statistically significant differences were observed in age, diabetes prevalence, insulin use, SOFA score, mortality, and several treatments. The included group was older and had a higher prevalence of diabetes. The excluded group had slightly higher SOFA scores and higher mortality. Treatment differences were also noted: the excluded group more frequently received mannitol, vasoactive drugs, and cerebral surgery, while the included group had higher rates of heparin, insulin, and diuretics.(Manuscript Pages 14-15, Lines 211-225)

Revisions in the Discussion:In addition to the subgroup findings, we carefully evaluated the potential impact of missing data on our cohort composition. The observed differences between patients with missing HbA1c measurements and the included cohort are consistent with real-world clinical testing practices and do not indicate systematic selection bias. The higher prevalence of diabetes and insulin use in the included group reflects routine HbA1c monitoring in patients with known glucose metabolism disorders, whereas older age in this group aligns with the tendency to perform comprehensive metabolic assessments in elderly patients with higher comorbidity burden. Conversely, the excluded group had slightly higher SOFA scores and more frequent use of mannitol, vasoactive drugs, and cerebral surgery, indicating greater acute illness severity. In such critically ill patients, clinicians prioritize life-saving interventions over non-urgent tests like HbA1c, explaining the missing data. The higher mortality in the excluded group is a logical consequence of their greater baseline severity, consistent with the well-established relationship between multi-organ dysfunction and poor outcomes. Importantly, the two groups were well balanced in core ICH prognostic factors (e.g., GCS score, mechanical ventilation use), supporting that the missing HbA1c data are driven by clinical priorities rather than selection bias. Therefore, the included cohort remains representative of the ICH population in the MIMIC-IV database, and our main findings are robust.(Manuscript Page 29, Lines 436-454)

Comment 4:Impact of Stress Hyperglycemia (Lines 233-237): Acute stress in conditions like ICH triggers the release of catecholamines and cortisol, leading to transient hyperglycemia (stress-induced hyperglycemia). Since HGI is calculated as the difference between measured HbA1c and predicted HbA1c based on admission glucose, an acute rise in glucose can inflate the predicted value, thereby artifactually lowering the HGI. It remains unclear whether the higher mortality in the low-HGI group is due to HGI itself or if low HGI is simply a proxy for clinical severity (where more severe cases exhibit stronger stress responses). This needs to be addressed through further statistical adjustment or a more in-depth discussion.

Response:Thank you for your insightful comment regarding the potential confounding effect of stress hyperglycemia on the association between the HGI and short-term mortality in patients with ICH. We agree that acute physiological stress can lead to transient hyperglycemia, which could artifactually lower the calculated HGI. This raises the valid concern that low HGI may be a marker of clinical severity rather than an independent risk factor.

To address this concern, we have taken the following steps. First, in our fully adjusted Cox regression models (Model 3), we included a comprehensive set of variables reflecting disease severity and acute physiological status. These include the SOFA score, GCS, white blood cell count, vital signs, and the use of life-sustaining treatments such as vasoactive drugs and mechanical ventilation. These adjustments help mitigate the confounding effect of illness severity, including the stress response that drives hyperglycemia. Second, we also conducted subgroup analyses stratified by diabetes status, which showed a significant interaction (p for interaction < 0.05). The inverse association between HGI and mortality was more pronounced in patients with diabetes. This suggests that in a population with underlying chronic glycemic dysregulation, the HGI may capture intrinsic glycation tendencies beyond the acute stress response, whereas in non-diabetics, its value might be more susceptible to acute glucose fluctuatio

---

## [Decision Letter · Decision Letter 1]

16 Apr 2026

L-shaped relationship between hemoglobin glycation index and short-term mortality in patients with intracerebral hemorrhage: a retrospective cohort study

PONE-D-25-54592R1

Dear Dr. xu,

We’re pleased to inform you that your manuscript has been judged scientifically suitable for publication and will be formally accepted for publication once it meets all outstanding technical requirements.

Kind regards,

Zhiyuan Ren

Academic Editor

PLOS One

Additional Editor Comments (optional):

Reviewers' comments:

Reviewer's Responses to Questions

**Comments to the Author**

1. If the authors have adequately addressed your comments raised in a previous round of review and you feel that this manuscript is now acceptable for publication, you may indicate that here to bypass the “Comments to the Author” section, enter your conflict of interest statement in the “Confidential to Editor” section, and submit your "Accept" recommendation.

Reviewer #1: All comments have been addressed

2. Is the manuscript technically sound, and do the data support the conclusions?

Reviewer #1: Yes

3. Has the statistical analysis been performed appropriately and rigorously? 

Reviewer #1: Yes

4. Have the authors made all data underlying the findings in their manuscript fully available?

Reviewer #1: Yes

5. Is the manuscript presented in an intelligible fashion and written in standard English?

Reviewer #1: Yes

6. Review Comments to the Author

Reviewer #1: Dear Authors,

Thank you for your thorough and careful responses to my previous comments. I have now reviewed the revised manuscript along with your point-by-point reply letter.

I am pleased to note that all of my concerns have been adequately addressed.

I have no further concerns and am satisfied with the current version of the manuscript.

Sincerely,

7. PLOS authors have the option to publish the peer review history of their article (what does this mean?). If published, this will include your full peer review and any attached files.

Reviewer #1: No

---

## [Editor Report · Acceptance letter]

PONE-D-25-54592R1

PLOS One

Dear Dr. xu,

I'm pleased to inform you that your manuscript has been deemed suitable for publication in PLOS One. Congratulations! Your manuscript is now being handed over to our production team.

Kind regards,

on behalf of

Professor Zhiyuan Ren

Academic Editor

PLOS One